

# Effects of $^{60}$Coγ-irradiation combined with sodium dehydrogenate on post-harvest preservation and physiological indices of *Volvariella volvacea*

Mingjuan Mao[1,2], Lin Ma[2], Ning Jiang[2], Jinsheng Lin[2], Shaoxuan Qu[2], Huiping Li[2], Ping Xu[2], Di Liu[1] and Lijuan Hou[2]

[1] Department of Horticulture and Landscape Architecture, College of Agriculture, Yanbian University, Yanji, Jilin, China
[2] Jiangsu Key Laboratory for Horticultural Crop Genetic Improvement, Institute of Vegetable Crop, Jiangsu Academy of Agricultural Sciences, Nanjing, Jiangsu, China

Corresponding authors
Di Liu, liudi@ybu.edu.cn
Lijuan Hou, mybailinggu@126.com

## ABSTRACT

*Volvariella volvacea* is a mushroom known for its high palatability and nutritional value. However, it is susceptible to spoilage thus making it challenging to preserve and keep fresh after harvest, resulting in constraints in long-distance transportation and long-term storage. This study aimed to investigate the feasibility of using irradiation and sodium dehydrogenate (SD) as a preservative in the preservation process of *V. volvacea*. The effects of three treatments of 0.8 kGy $^{60}$Coγ irradiation (B), 0.04% SD (C), combined with 0.04% SD and 0.8 kGy $^{60}$Coγ irradiation (A) on the postharvest freshness of *V. volvacea* were investigated. The assessment indices for *V. volvacea*, including appearance, browning rate, weight loss, respiration rate, MDA content, antioxidant enzyme activities, vitamin C (Vc), and soluble protein content, were measured and compared. The three treatments were compared to determine the changes in storage time over 7 days post-harvest. The results demonstrated that the hardness of the fruiting body exhibited a significant increase of 81.19%, 97.96% and 168.81% in comparison to the control, B and C, respectively, following the application of the treatment A. Compared to the control group, the soluble protein content was significantly increased by 20.28%. Respiration intensity and browning rate were significantly lower in the control treatment, decreasing by 35.07% and 45.49% respectively. On the 6th day of storage, the activities of SOD and POD increased by 81.06% and 73.71%, respectively, compared to the control, which significantly delayed the senescence of the fruiting bodies. The Vc content was significantly increased by 50.27%, 133.90%, and 101.39% in treatment B, which received 0.8 kGy $^{60}$Coγ irradiation alone, compared to the control, treatment A, and treatment C, respectively. The treatment C alone significantly reduced respiratory intensity and MDA variables by 39.55% and 31.01%, respectively, compared to the control. The findings can provide theoretical references and technical support for extending the preservation period of *V. volvacea* after harvesting by using irradiation and sodium dehydrogenate as a preservative.

# INTRODUCTION

*Volvariella volvacea*, also known as the Chinese mushroom, is one of the exported mushrooms in China (*Gong et al., 2022*). It is a tasty nutritionally and medicinally valuable mushroom (*Zhu et al., 2019*), with great application value in healthcare and medical treatment (*Wu et al., 2023*). *V. volvacea* veil opening 3–5 h after harvesting, are easily autolytic, have a shelf life of 1–2 d (*Zhao, 2022*), and the process is always in a state of water loss, which reduces the quality of *V. volvacea* and loses their commercial value. Edible mushrooms undergo certain regular changes in physicochemical properties after harvesting, such as changes in tissue composition, respiratory metabolism, energy metabolism (*Li et al., 2017*). Low-temperature storage (0–5 °C) is one of the most effective methods of preserving *Pleurotus ostreatus*, *Lentinula edodes* and other mushrooms, and the optimum storage temperature for *V. volvacea* is 15 °C (*Wang et al., 2021*). *V. volvacea* is known for its susceptibility to temperature variations, especially in terms of the fragility of its fruiting body towards frostbite. In addition, the short shelf life and unfavorable storage conditions contribute to the decreased market competitiveness of *V. volvacea* products. Consequently, the preservation dilemma emerges as a fundamental barrier limiting the production of *V. volvacea*, while also serving as a crucial factor essential in addressing concerns related to long-distance transportation (*Qiu et al., 2022*).

At present, some research undertakings at home and abroad have conducted research on the mechanism of quality deterioration of *V. volvacea* and developed storage and preservation strategies (*Shao, 2023*). It is known that the storage and preservation methods of *V. volvacea* mainly include air-conditioning treatment (*Wang et al., 2017*), irradiation preservation (*Shi, 2020*), ultrasonic combined with humidity preservation (*Huang, 2017*), biopreservation technology and chemical preservation (*Lin et al., 2024*). The traditional single preservation process has limitations, prompting the continuous development and improvement of preservation processes for *V. volvacea* (*Xu et al., 2024*). At present, irradiation treatment is widely used in the field of food sterilization and preservation. Commonly used sources of irradiation include [60]Co-γ, ultraviolet light, electron beam and X-rays, with [60]Co-γ irradiation being the most widely used in the production of practical species (*Jin & Dong, 2017*). The principle is to eliminate or reduce microbial survival to prevent accelerated spoilage, as well as to inhibit or delay the physiological and biochemical responses of the mushroom, thus reducing the rate of weight loss, respiration rate and opening (*Saliha et al., 2022*). *V. volvacea* can continue to grow for a period of time after harvest. And preservation is essential to delay the process of ripening and aging of *V. volvacea* (*Li et al., 2019*).

Chemical preservation techniques have been reported for *Flammulina velutipes* (*Zhang, 2021*), *Lentinula edodes* (*Li et al., 2023*), *Agaricus bisporus* (*Seyed et al., 2023*) and *Hypsizygus marmoreus* (*Zhao, Mu & Li, 2016*), but the use of food-grade sodium dehydrogenate (SD) for the preservation and conservation of edible mushrooms remains

obscure. SD is a broad-spectrum, highly bacteriostatic food preservative with strong growth inhibition of moulds, yeasts and bacteria, and has been approved by FAO/WHO for use in Europe and the United States for many years (*Li et al., 2004*). In China, the "Standard for the Use of Food Additives" (GB2760-2014) stipulates that the general usage range of 0.3 to 0.5 g/Kg. Furthermore, it was noted that these additives can be degraded in the human body during the metabolic process. It can be degraded to acetic acid during metabolism, which is non-toxic to human body, and has been widely used in the preservation of soy sauce, fruit juice, bread, noodles (*Yang et al., 2021*), strawberries (*Abdallah et al., 2023*), Water Chestnut (*Li et al., 2022*), Blueberry (*Wu, Qin & Yang, 2022*), *etc.*, and has achieved good preservation effect in prolonging shelf life and improving quality. While there is limited documentation of its use in the preservation of *V. volvacea*, it has nonetheless produced a successful preservation result that has not been previously reported.

Based on our previous research, [60]Coγ irradiation and SD treatment of *V. volvacea* have prolonged the shelf life of *V. volvacea* to different degrees, but the research on the effect of the combination of the above two methods on the preservation of *V. volvacea* is still unknown.

The objective of this study was to investigate the impact of gamma radiation and sodium acetate immersion treatments on the quality of *V. volvacea* during storage. Furthermore, the study sought to establish a theoretical foundation for further investigation into the preservation of *V. volvacea* and to prolong its postharvest shelf life.

## MATERIALS AND METHODS

### Materials

The fruiting bodies were transported to the irradiation center of Ruidisheng Technology Company (Jiangsu, Nanjing, China). Plastic boxes containing mushroom trays (110 ± 10 g) were placed on a conveyor and irradiated with [60]Co gamma radiation of 0.8 kGy. In the irradiation center (the activity was $1.41 \times 1,014$) of [60]Co irradiation device. After irradiation, the samples were stored at 16 ± 0.5 °C and 55% relative humidity (RH), and data were recorded for further analysis.

The main reagents were SD (food grade), sodium dihydrogen phosphate, disodium hydrogen phosphate, polyvinylpyrrolidone and analytical grade catechol. The malondialdehyde (MDA) assay kit, peroxidase (POD) assay kit and superoxide dismutase (SOD) assay kit were purchased from Nanjing Jianjian Technology [60]Co.

### Instruments and equipment

The instruments used for the analysis include; 64R freezing centrifuge (Beckman, Brea, CA, USA); HH-6 digital thermostatic water bath (Guohua Electric Appliance Co., Ltd., Changzhou, Jiangsu, China); Respirometer (PBIDansensor, Ringsted, Denmark); Enzyme labelling device (Shanghai Precision Scientific Instrument Co., Ltd., Shanghai, China); Film sealing machine (Lianyungang Microwave Electric Apparatus Factory, Jiangsu Province, Lianyungang, China); FA2004A, JA6102 electronic scales (Shanghai Jingtian

**Table 1 Sensory evaluation indices of *V. volvacea* and their scores.**

| Sensory evaluation | Score and description | | | |
|---|---|---|---|---|
| | 1 | 2 | 3 | 4 |
| Odor | No smell of fresh | Slight smell fresh | Mushroom flavor | Typical mushroom flavor |
| Color | Internal turn yellow | Yellow | A slight browning | No browning |
| Firmness | Extreme softening | Change softening | Hard very hard | Firmness |
| Veil opening | Veil opening have water stain on | Rupture of membranes | Crack | No veil opening |
| Rotting decay | The top of mushroom | Have water stain on the bottom of mushroom | Shrikage | No rotting decay |

Electronic Instrument Factory, Shanghai, China); Texture Analyzer-TA.XT2i (Shanghai Ruiyan International Trade Co., Ltd., Shanghai, China).

## Experimental design

There were three treatments for *V. volvacea*: (i) Compound SD treatment and $^{60}$Coγ irradiation: the combined conclusions of the two were that 0.04% SD solution was chosen to soak the mushrooms for 45 s and then irradiate the mushrooms with 0.8 kGy $^{60}$Coγ irradiation (hereinafter referred to as method A); (ii) $^{60}$Coγ irradiation treatment (*Hou et al., 2018*) (hereinafter referred to as method B); (iii) SD treatment (*Hou et al., 2014*) (hereinafter referred to as method C); the control group is the sub-entity without any treatment.

The fruiting bodies were transported to the irradiation center of Ruidisheng Technology Company (Jiangsu Nanjing, China). Plastic boxes containing mushroom trays (110 ± 10 g) were placed on a conveyor and irradiated with $^{60}$Co gamma radiation of 0.8 kGy. In the irradiation center (the activity was 1.41 × 1,014) of $^{60}$Co irradiation device. Upon irradiation midway through the process, the packaging box changed in appearance due to irradiation, facilitated by the use of dichromate dosimeters to monitor the absorbed dose of the products, ensuring an uptake of 0.8 kGy. Post-irradiation, the samples were stored under controlled conditions at 16 ± 0.5 °C and 55% relative humidity (RH), with data meticulously recorded for subsequent analysis. To assess the effects of the treatment, three boxes from each group were selected at 24-h intervals, with one replicate per box. The mushrooms were then subjected to organoleptic evaluation and physiological and biochemical assessments. Each group was treated with three biological replicates.

## Determination of sensory evaluation indices

The sensory evaluation was mainly based on the method of *Hou et al. (2018)*, which was carried out by five experienced panelists from the Department of Food Science and Technology, Nanjing Agricultural University, China, to evaluate the flavour, surface colour, browning, hardness, openness and degree of decay of the sample mushrooms. Each evaluation index was divided into four levels corresponding to four scores for a total of five evaluation indices with a total score of 20 (Table 1). The texture index of *V. volvacea* was determined by tactile assessment using the thumb and forefinger, while the degree of

browning, opening, and rot was evaluated through visual inspection. This evaluation method involved scoring each index three times and monitoring continuously for 5 days to document the final score.

## Mushroom hardness

The hardness measurement was conducted according to the method of *Li (2018)*, and the hardness change of the fruiting body of *Volvariella volvacea* was determined by the food physical property tester TA-XT2i. Among them, the probe diameter is 2 mm, the downward pressure distance is 5 mm, the inertial force is 5 g, the speed before measurement is 2 mm/s, the speed in detection is 2 mm/s, and the speed after detection is 10 mm/s. The hardness unit was N, and the maximum peak value was used to represent the hardness index. Five samples were randomly selected from each group, and six points were selected symmetrically and evenly for each sample. Finally, the average value was computed.

## Mushroom browning rate

The browning rate was determined according to the method of *Li (2018)*. Absorbance was measured at 450 nm and the browning rate of *V. volvacea* was determined by categorizing the numerical data.

## Weight loss

According to *Wang et al. (2022)*, the formula of mushroom mass weight loss was computed as follows; (mushroom mass before storage-mushroom mass after storage)/ mushroom mass before storage × 100 = mass loss rate (%).

## Respiration rate

Static measurements at a storage temperature of 4 ± 1 °C were used, and respiration rates were calculated from oxalic acid consumption after titration of sodium hydroxide ($C_2H_2O_4$, 0.2 mol/L) with oxalic acid (*Li, 2018*).

## SOD and POD enzyme activities and MDA content assay

Differently treated *V.volvacea* tissues (1.0 g) were homogeniszeed with $K_3PO_4$ buffer (pH 6.8), and the supernatant was collected after centrifugation for 10 min at 10,000 × gand 4 °C for the determination of SOD and POD enzyme activities and MDA content assays. The results were analyszed using commercial assay kits (Nanjing Jianjian Bioengineering Institute, Nanjing, Jiangsu, China).

## Determination of Vc and soluble protein content

The Vc content was determined by the 2,6-dichlorophenol indophenol titration method, the soluble protein content was determined by the method of *Shao et al. (2022)*.

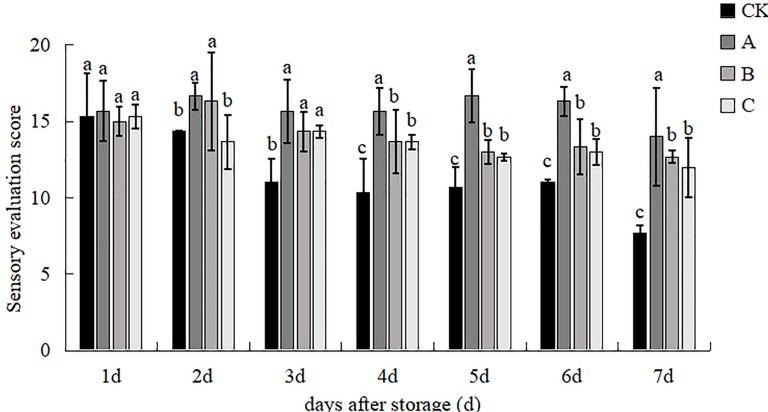

**Figure 1** **Sensory scores of *V. volvacea* in different treatments.** (CK) Sub-entity without any treatment; (A) Mushrooms were soaked in 0.04% SD solution for 45 s, and then irradiated with 0.8 kGy 60Coγ; (B) 60Coγ irradiation treatment; (C) 0.04% SD solution soaked mushrooms for 45 s. The values represent the means ± standard deviations. Statistical analysis was performed using t-test. The lowercase letters indicate a significant difference at the 0.05 level, and the same letters indicate insignificant differences, as below.

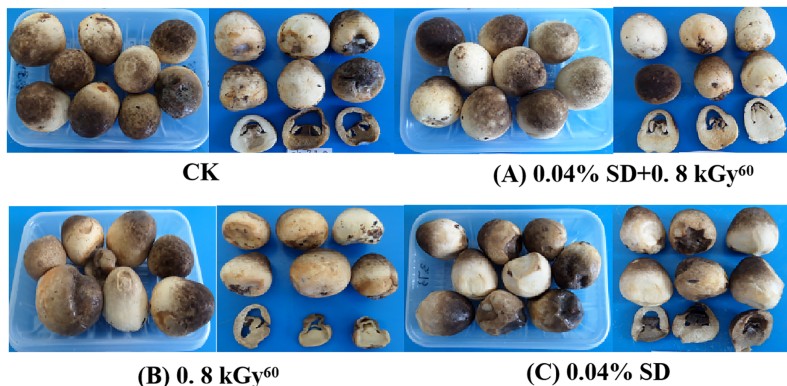

**Figure 2** **Effect of different treatments on the appearance quality of *V. volvacea* fruit bodies.**

## Statistical analysis

All data were expressed as mean ± SD. Results were analyzed for the significance of differences using SPSS 22.0 software (SPSS, Inc., Chicago, IL, USA), differences between groups were analyzed using Duncan's multiple comparisons and plotted using Excel 2017 and Origin 2021 software. Differences between means were considered significant at $P < 0.05$.

## RESULTS

### Sensory evaluation

The sensory evaluation of the mushrooms from the different treatments during the storage period is shown in Figs. 1 and 2. The sensory scores of the mushrooms from all treatments showed a decreasing trend during the 7-day storage period, and the overall score of

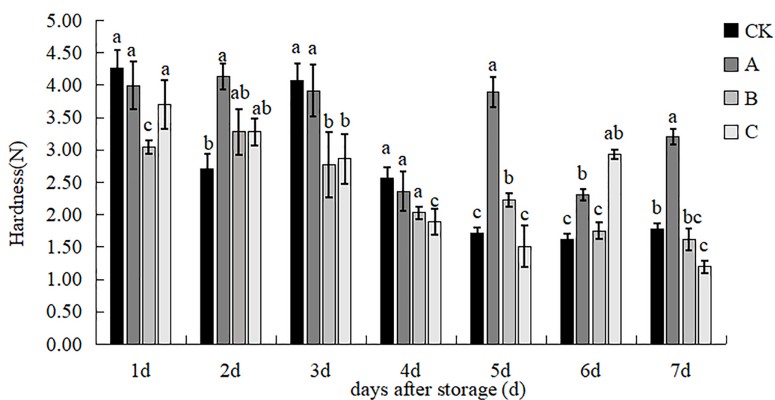

**Figure 3** The effect of different treatment on the hardness of the *V. volvacea*. (CK) Sub-entity without any treatment; (A) Mushrooms were soaked in 0.04% SD solution for 45 s, and then irradiated with 0.8 kGy 60Coγ; (B) 60Coγ irradiation treatment; (C) 0.04% SD solution soaked mushrooms for 45 s. The values represent the means ± standard deviations. Statistical analysis was performed using t-test. The lowercase letters indicate a significant difference at the 0.05 level, and the same letters indicate insignificant differences.

treatment A was higher than that of the other treatments. At the onset of the storage period, there were no discernible variations between the treatments. However on day 2, treatment A recorded the highest score on t, with treatment B following closely behind. All three treatments were significantly different from control, and the score of treatment A was 42.45% higher than that of the control group, but there was no significance among the three treatments on d 3. From d 4 to d 7, there were significant differences between treatment A and all other treatments and between treatments B and C and the control group, and from day 4 onwards, treatment A was 51.69%, 53.05, 48.45% and 82.53% higher than the control, respectively. On the 7th day of the storage period, the browning and opening degree of *V. volvacea* of treatment A was significantly lower than the other treatments after mushroom cross-cutting, and the hardness was better than the other treatments (Fig. 2). The degree of decay, shrinkage and browning of the B treatment was also higher and the integrity of the mushroom was lower; the degree of opening and decay of the C treatment was better than that of the control and B treatments, but browning and hardness were weaker than that of the A treatment, so that the appearance of the mushrooms in the C treatment was superior to that of the B and control.

## Hardness

The effects of the different treatments on the hardness of the straw mushroom are shown in Fig. 3. Treatment B exhibited a notably reduced mushroom hardness compared to the other treatments on day 1 of the storage period. On day 2, treatment A displayed a significant variance in comparison to the control group, with the mushroom hardness being notably higher in treatment A and the control groups than in treatments B and C on day 3. Treatment C had the lowest hardness, with all other treatments significantly surpassing treatment C on day 4. The hardness of treatment A surpassed that of the control, treatment B, and treatment C after 5 days, showing a significant increase of 127.19%, 74.21%, and 158.13%, respectively. By day 6, treatments A and C exhibited

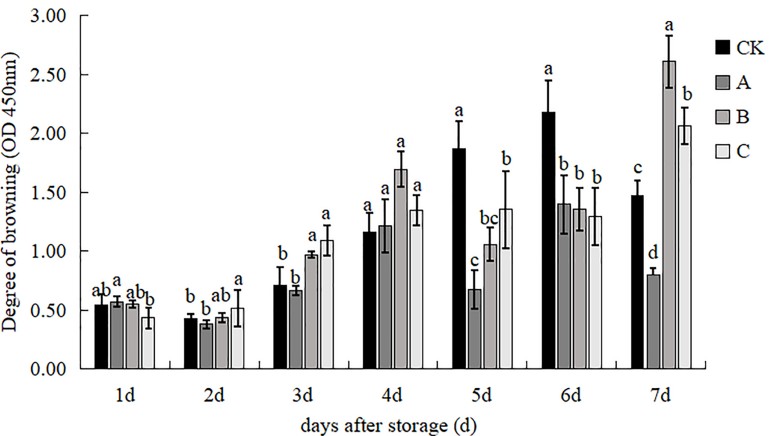

**Figure 4** **The effect of different treatment on the degree of browning of *V. volvacea*.** (CK) Sub-entity without any treatment; (A) Mushrooms were soaked in 0.04% SD solution for 45 s, and then irradiated with 0.8 kGy 60Coγ; (B) 60Coγ irradiation treatment; (C) 0.04% SD solution soaked mushrooms for 45 s. The values represent the means ± standard deviations. Statistical analysis was performed using t-test. The lowercase letters indicate a significant difference at the 0.05 level, and the same letters indicate insignificant differences.               

significantly higher levels of hardness compared to the control and B treatment. However, the difference in hardness between treatments A and C was not statistically significant until day 7. The hardness of fruiting bodies in treatment A was notably higher than that of the other treatments, showing an increase of 81.19%, 97.96%, and 168.81% compared to the control, B, and C, respectively. Throughout the storage period, the hardness of mushrooms in treatment A remained consistently high, which helped prevent softening. Treatments C and B showed similar levels of hardness, while the control group exhibited a linear decrease in hardness from the 4th to the 7th day, significantly lower than the other treatments, indicating noticeable softening of the mushrooms.

## Degree of browning

The impact of various treatments on the discoloration of *V. volvacea* is illustrated in Fig. 4. In the initial two days of storage, there was no significant difference in the rate of discoloration among the treatments. However, by the third day, treatment B and treatment C exhibited significantly higher rates of discoloration compared to the control group and treatment A. The discoloration process accelerated on the fourth day, yet there was no significant variance observed between the treatments. The browning rate of the control was significantly higher than that of the other treatments, while treatment A had the lowest browning rate, 63.98% lower than the control, and the difference was significant at d 5. The control was still significantly higher than that of the other treatments, and the difference between the three experimental treatments was not significant at d 6. By day 7, the browning rate of the control group decreased, whereas treatment B showed a notably higher browning rate than the other treatments. In contrast, treatment A exhibited a significantly lower browning rate than the other treatments, with a reduction of 45.49% compared to the control, 61.22% compared to treatment B, and 61.05% compared to

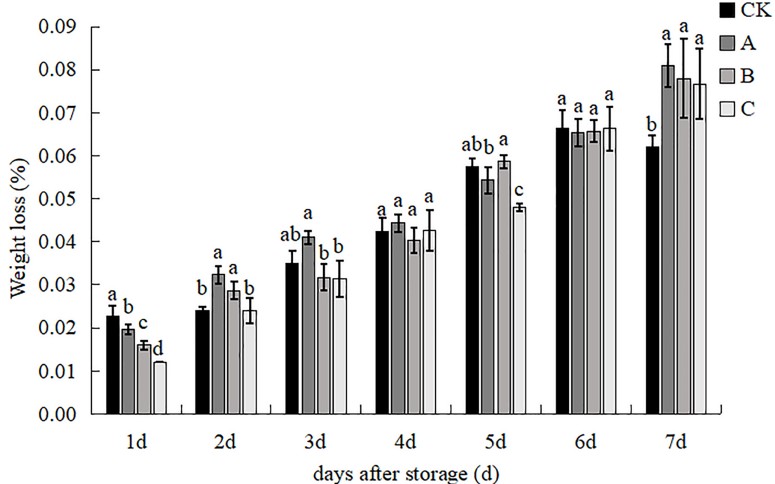

**Figure 5 The impact of different treatment on the weight rate of *V. volvacea*.** (CK) Sub-entity without any treatment; (A) Mushrooms were soaked in 0.04% SD solution for 45 s, and then irradiated with 0.8 kGy 60Coγ; (B) 60Coγ irradiation treatment; (C) 0.04% SD solution soaked mushrooms for 45 s. The values represent the means ± standard deviations. Statistical analysis was performed using t-test. The lowercase letters indicate a significant difference at the 0.05 level, and the same letters indicate insignificant differences.

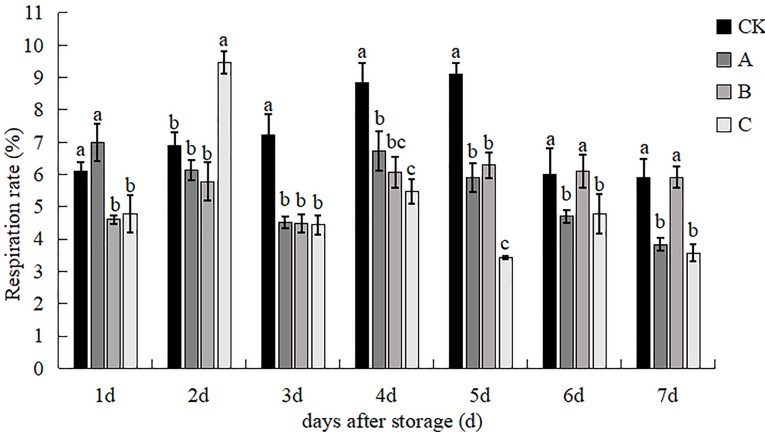

**Figure 6 The effect of different treatments on the breathing rate of *V. volvacea*.** (CK) Sub-entity without any treatment; (A) Mushrooms were soaked in 0.04% SD solution for 45 s, and then irradiated with 0.8 kGy 60Coγ; (B) 60Coγ irradiation treatment; (C) 0.04% SD solution soaked mushrooms for 45 s. The values represent the means ± standard deviations. Statistical analysis was performed using t-test. The lowercase letters indicate a significant difference at the 0.05 level, and the same letters indicate insignificant differences.

treatment C. These findings indicate that treatment A is more effective in reducing browning in *V. volvacea* making it a preferable preservation technique.

## Weight loss

The effect of the different treatments on the rate of weight loss of the straw mushroom is shown in Fig. 5. The weight loss rate of the *V. volvacea* under each treatment showed an increasing trend with increasing days of storage. There were significant differences

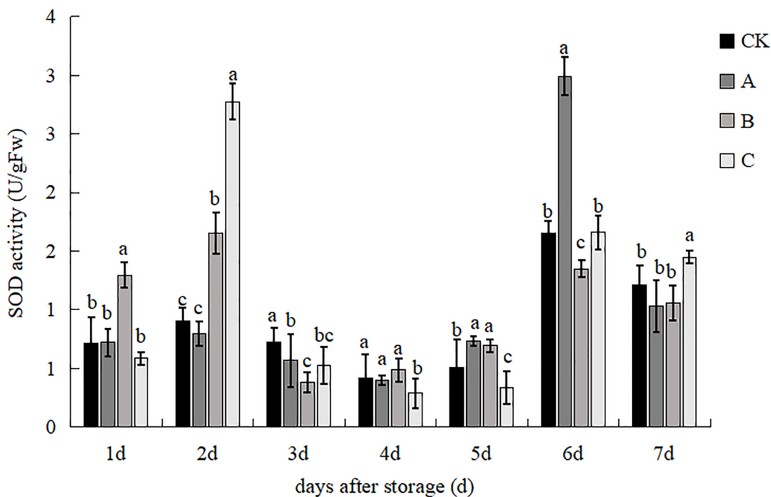

**Figure 7 The effect of different treatments on the SOD activity of *V. volvacea*.** (CK) Sub-entity without any treatment; (A) Mushrooms were soaked in 0.04% SD solution for 45 s, and then irradiated with 0.8 kGy 60Coγ; (B) 60Coγ irradiation treatment; (C) 0.04% SD solution soaked mushrooms for 45 s. The values represent the means ± standard deviations. Statistical analysis was performed using t-test. The lowercase letters indicate a significant difference at the 0.05 level, and the same letters indicate insignificant differences.

between treatments and the highest weight loss was found in the control group, with treatments A, B and C being lower than the control by 13.04%, 30.43% and 47.83% respectively on d 1. In the 2nd and 3rd day, the weight loss rate of treatment C remained low, in the 4th and 6th day, the weight loss rate of each treatment continued to increase, but there was no significant difference between the treatments. The weight loss rate of the control group and treatment B was significantly higher than that of treatments A and C, and the weight loss rate of treatment C was the lowest, followed by that of treatment A. The weight loss rate of treatments C and A compared to the control group was 15.79% and 5.26%, respectively and 79% and 5.26%, respectively, in the 5 d. By the 7th day, the weight loss rate of the control group was significantly lower than that of the other 3 treatments, and the weight loss rate of group C was reduced by 13.04%, 30.43% and 47.83%, respectively, compared to the control. Only the control group was significantly lower than the other three treatments, and by combining the 7-day data of each treatment, the C treatment method was able to maintain a lower rate of weight loss.

## Respiration rate

The effects of different treatments on the respiration rate of *V. volvacea* and the time of peak respiration rate were different (Fig. 6). With the increase of storage time, the respiration rate of the control group (except the 2nd day) was significantly higher than that of the other treatments, and the respiration rate of treatment A showed a gradual decreasing tendency from the 4th to the 7th day. The respiration rate of the control group was significantly higher than that of the B and C treatments on day 1. The respiration rate of treatment A was markedly elevated in contrast to treatments B and C, while no significant disparity was noted when compared to the control group on the 2nd day. The

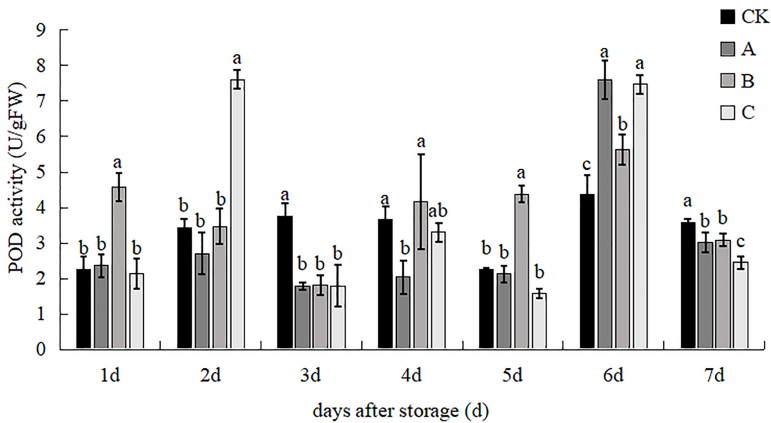

**Figure 8 The impact of different treatment on the active POD activity of *V. volvacea*.** (CK) Sub-entity without any treatment; (A) Mushrooms were soaked in 0.04% SD solution for 45 s, and then irradiated with 0.8 kGy 60Coγ; (B) 60Coγ irradiation treatment; (C) 0.04% SD solution soaked mushrooms for 45 s. The values represent the means ± standard deviations. Statistical analysis was performed using t-test. The lowercase letters indicate a significant difference at the 0.05 level, and the same letters indicate insignificant differences.

respiration rate of treatment C was the highest at 9.47%, which was significantly different from that of the other three treatments on the 3rd day. The respiration rate of the control group was the highest and the respiration rate of the C treatment was the lowest, and the respiration rate of the C treatment was 38.01% lower than that of the control group and treatment A and 18.48% lower than that of treatment A after 4th day. The control group had the highest respiration rate and the C treatment had the lowest respiration rate and differed from the other treatments. Treatment C showed reduced respiration by 62.47%, 42.00% and 45.64% compared to the control, A and B treatments, respectively, after the 5th day. On days 6 and 7, the differences between the control and B treatments were not significant but were all significantly higher than the A and C treatments. The respiration rate of treatment C was reduced by 39.55% compared with that of the control group, and the respiration rate of treatment A was reduced by 35.07% compared with that of the control group on the 7th day, which indicated that A and C treatments could effectively reduce the respiration rate so that *V. volvacea* could be preserved for a longer time.

## SOD activity

As shown in Fig. 7, treatment B had a significantly higher SOD enzyme activity than the other three treatments, while the difference between the other three treatments was not significant on day 1. Treatment C had a significantly higher SOD enzyme activity than the other treatments, with values 207.53%, 248.43% and 68.00% higher than the control, A and B treatments, respectively. The difference between the control and A treatments was not significant on day 2. The SOD enzyme activity of the control group was significantly higher than that of the three treatment groups. Treatment B had the lowest activity and was significantly different from the other treatments on day 3. Treatment C had significantly lower SOD enzyme activity than the other treatments on 4th d, but there was no significant difference between the other three treatments. Treatment A had the highest SOD enzyme

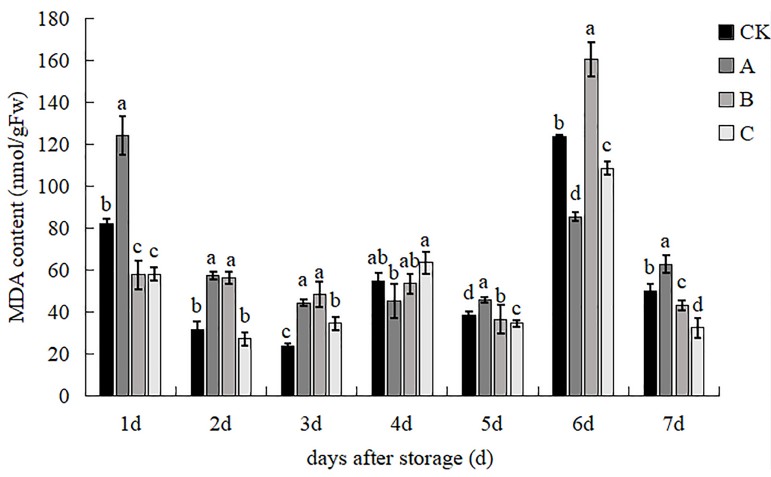

**Figure 9 The impact of different treatment on the MDA content of *V. volvacea*.** (CK) Sub-entity without any treatment; (A) Mushrooms were soaked in 0.04% SD solution for 45 s, and then irradiated with 0.8 kGy 60Coγ; (B) 60Coγ irradiation treatment; (C) 0.04% SD solution soaked mushrooms for 45 s. The values represent the means ± standard deviations. Statistical analysis was performed using t-test. The lowercase letters indicate a significant difference at the 0.05 level, and the same letters indicate insignificant differences.

activity, which was significantly different from treatments control and C treatment, but not from treatment B on day 5. The SOD enzyme activity of treatment A was significantly higher than that of the other treatments. SOD enzyme activity increased treatment A by 81.06%, 121.70% and 80.30% compared to the control, B and C treatments, respectively at day 6. The difference between control and treatment B was significant, but the difference with treatment C was not significant. The enzyme activity was reduced in all the treatments, with treatment C showing a significant increase on day 7 in comparison with the other treatments. However, there was no significant difference between control, A and B treatments. Overall, the combination treatment A increased SOD enzyme activity and was more favourable for mushroom preservation.

## POD activity

The effects of different treatments on the POD enzyme activity of *V. volvacea* are shown in Fig. 8. After 1 d storage, the POD enzyme activity of treatment B was significantly higher than that of the other treatments and there was no significant difference between the other three treatments, which was 102.75% higher than that of the control. After 2 d, the POD enzyme activity of treatment C increased sharply and was significantly higher than that of the three treatments of control, A and B, which was 121.46%, 181.22% and 119.22% higher than that of the control, A and B treatments, respectively, and there was no significant difference between these three treatments. The POD enzyme activity of the control was significantly higher than that of the three treatments, increasing by 219.17%, 107.79% and 109.99% than those of A, B and C, respectively, but there was no significant difference between treatments A, B and C on day 3. The POD enzyme activity of treatment B was the highest, but it had a significant difference only with treatment A on day 4. POD enzyme activity of treatment B was significantly increased, POD enzyme activity of treatments

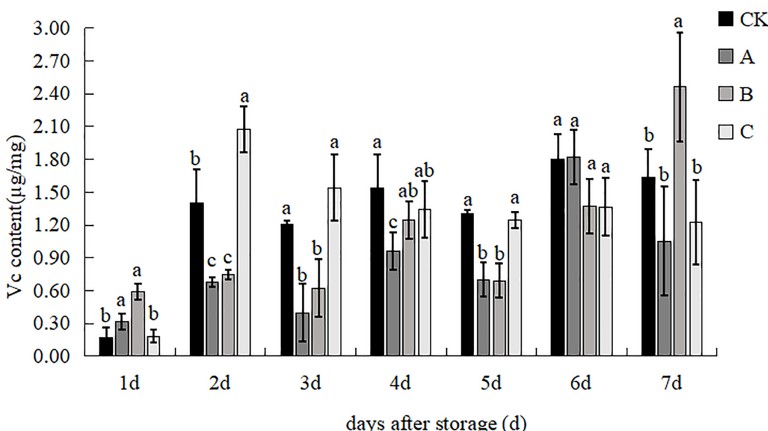

**Figure 10 The effect of different treatment on the content of Vc in the *V. volvacea*.** (CK) Sub-entity without any treatment; (A) Mushrooms were soaked in 0.04% SD solution for 45 s, and then irradiated with 0.8 kGy 60Coγ; (B) 60Coγ irradiation treatment; (C) 0.04% SD solution soaked mushrooms for 45 s. The values represent the means ± standard deviations. Statistical analysis was performed using t-test. The lowercase letters indicate a significant difference at the 0.05 level, and the same letters indicate insignificant differences.

control, A and C were significantly increased by 95.50%, 105.87% and 179.17%, respectively on day 5. POD enzyme activity of all treatments increased and treatment A had highest activity which was 73. 71% and 34.91% higher than the control and B treatments, respectively, on day 6. The highest POD enzyme activity was found in the control and the lowest in treatment C. The control group increased by 19.17%, 16.46% and 46.47% compared to treatments A, B and C, respectively, on day 7.

## MDA content

The effect of different treatments on the MDA content of *V. volvacea* is shown in Fig. 9. Treatment B had the lowest MDA content, which was 29.82%, 53.64% and 0.71% lower than control, A and C treatments, respectively on day 1. Treatment C had significantly lower MDA than treatments A and B, and the difference with the control was not significant on day 2. All three treatment groups were significantly higher than the control group on d 3, treatment A had the lowest and treatment C the highest, with A being 28.79% lower than C on d 4. Treatment C was 9.59%, 24.11% and 5.02% lower than control, A and B, respectively, on the 5th day. The MDA content of the four treatments increased sharply, with the increase in treatment B being the greatest and significantly different from the other treatments, while the MDA content of treatment A was the lowest and was 46.76% lower than that of treatment B on day 6. The MDA content of all treatments was lower on day 7 than on day 6, and treatments B and C were significantly lower than the control, with reductions of 12.38% and 31.01%, respectively, compared to the control. The results indicate that each treatment had the highest degree of membrane lipid peroxidation on day 6 of storage and that both irradiation and sodium acetate treatments alone significantly reduced the degree of lipid peroxidation of cell membranes.

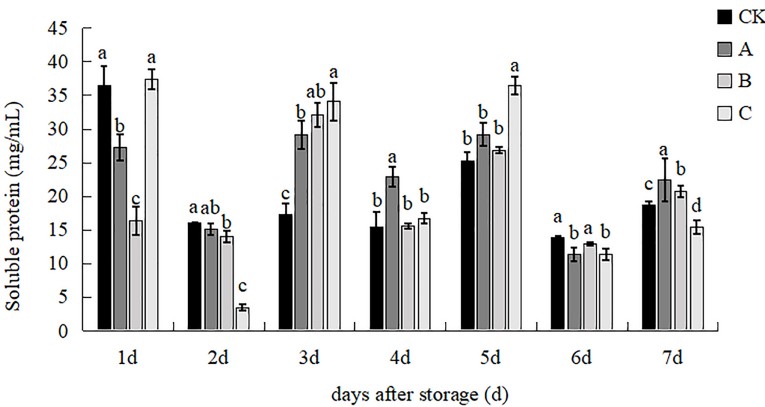

**Figure 11 The effect of different treatment on the soluble protein of the *V. volvacea*.** (CK) Sub-entity without any treatment; (A) Mushrooms were soaked in 0.04% SD solution for 45 s, and then irradiated with 0.8 kGy 60Coγ; (B) 60Coγ irradiation treatment; (C) 0.04% SD solution soaked mushrooms for 45 s. The values represent the means ± standard deviations. Statistical analysis was performed using t-test. The lowercase letters indicate a significant difference at the 0.05 level, and the same letters indicate insignificant differences.

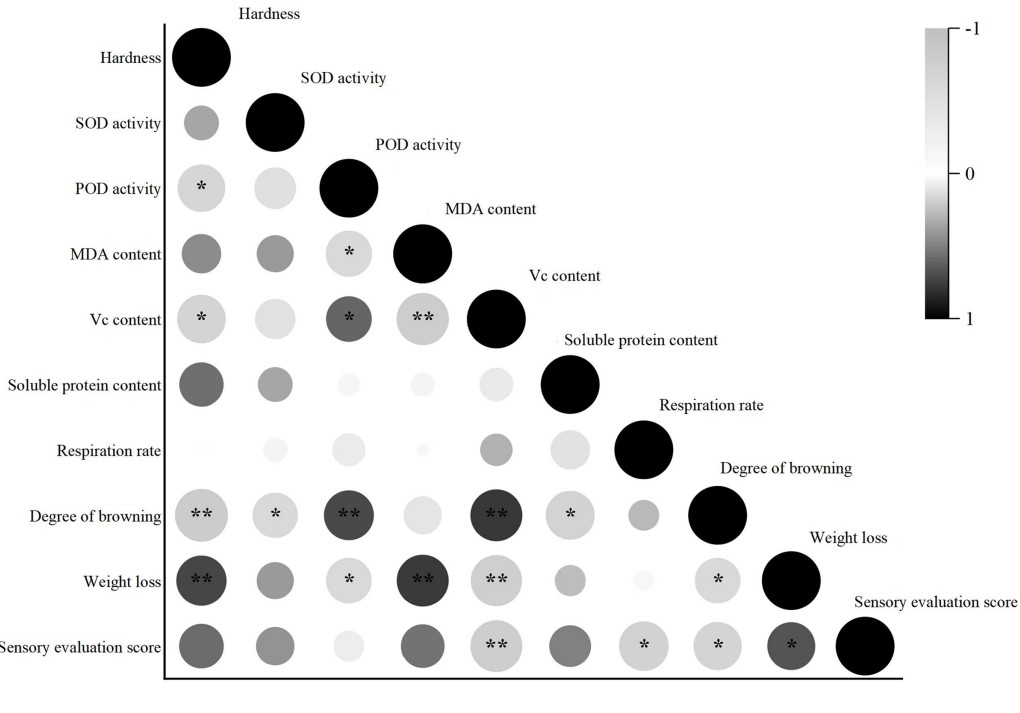

* p<=0.05  ** p<=0.01

**Figure 12 Correlation analysis of different indicators of *V. volvacea*.**

## Vc content

The effects of different treatments on the Vc content of *V. volvacea* are shown in Fig. 10. The Vc content of treatment B was significantly higher than that of the control group and treatment C, but the difference with treatment A was not significant at d 1. The Vc content

of treatment C was significantly higher than that of the other treatments, increasing by 48.39%, 26.20% and 177.14% at d 2. The content of treatment C was significantly higher than that of treatments A and B, and there was no significant difference with that of the control group at day 3. The Vc content was highest in the control group and lowest in treatment A, with a 1.61 times difference observed at day 4. The control group was significantly higher than that of treatments A and B, and there was no significant difference with that of the control group at d 5. There was no significant difference between treatments at d 6. The Vc content of treatment B increased sharply to 2.46 μg/mg and was significantly different from all other treatments, which was 50.27%, 133.90% and 101.39% higher than that of control, A and C treatments at d 7.

## Soluble proteins

As shown in Fig. 11, the content of treatment C was significantly higher than that of treatments A and B, and there was no significant difference with that of the control at d 3. The control group had the highest Vc content while treatment A had the lowest, and the difference was 1.61 times at d 4. The content of all treatments increased, and treatment C had the highest content, which was significantly higher than the control with 97.21% at d 3, treatment A had the highest content, with a significant increase of 48.78%, 47.17% and 37.41% over treatments control, B and C respectively at d 4. Treatment C had the highest content, with an increase of 44.11%, 24.80 and 35.71% over the control, A and B respectively, treatment B had the highest content and the differences between the other three treatments were not significant at d 6. Treatment A had the highest soluble protein content, followed by treatment B at d 7. The results showed that the combined treatment method was significantly higher than control with 97.21%. The results showed that the combined treatment method was more conducive to the accumulation of soluble protein in *V.volvacea* and more conducive to maintaining the quality.

## Correlation analysis

Pearson's coefficient was used to determine the correlation between the different indices measured on the *V. volvacea* at the 7th day of storage in this study (Fig. 12). SOD enzyme activity was significantly negatively correlated with browning rate ($P < 0.05$), and there was no significant correlation with any of the other treatments. POD activity was significantly positively correlated with Vc content (0.61), highly significantly positively correlated with browning rate (0.72), significantly negatively correlated with MDA content and weight loss rate, and significantly negatively correlated with respiration rate. There was a significant positive correlation between MDA and weight loss (0.77) and a significant negative correlation with Vc (−0.79). The Vc content demonstrated a highly significant positive correlation (0.78) with the browning rate and a highly significant negative correlation with weight loss and sensory evaluation. Soluble protein exhibited a significant negative correlation with the browning rate. Notable inverse correlation was observed between respiration rate and sensory evaluation, as well as between browning rate and weight loss rate and sensory evaluation. Significant negative correlation was observed between hardness and POD activity and Vc content, with a particularly strong negative correlation

evident between these variables and browning rate (−0.80). The indicators measured together showed that the indicators affecting the quality of the fruiting bodies (MDA content, browning rate and respiration rate) were negatively correlated with those favouring the preservation of the fruiting bodies (hardness, antioxidant enzyme activity, Vc and soluble protein content).

## DISCUSSION

In this study intothe screening of $^{60}$Co-γ irradiation dose, we established that 0.8 kGy of $^{60}$Co-γ was suitable for the preservation of *V. volvacea*, and this irradiation dose was consistent with that reported by *Zhou et al. (2018)*. *Hou et al. (2018)* reported that the microbial populations were lower than the control after 0.8 kGy $^{60}$Co-γ irradiation. However, they also observed that that the SOD enzyme activity was significantly higher than the control after 0.8 kGy $^{60}$Co-γ irradiation. In this study, we also found that irradiation treatment could significantly decrease respiration rate, reduce membrane lipid damage, significantly increase Vc content, and improve POD enzyme activity, which was the same as the dose investigated by *Shi (2020)*. It could decrease opening rate, weight loss, and conductivity of *V. volvacea*, as well as reduce membrane lipid damage, and prolong the freshness period by 3–6 d. At the same time, it was the same as that used by *Zhou et al. (2018)*, involving the application of 2 kGy $^{60}$Co-γ irradiation to inhibit tissue degeneration in shiitake mushrooms. Irradiation can inhibit the softening of mushroom tissues, thus ensuring that the cells are not destroyed to achieve the effect of preservation, its biological characteristics of the use of the appropriate irradiation dose (*Zhou et al., 2019*). Inappropriate irradiation dose will make the mushroom tissues produce excessive free radicals (*Zhang et al., 2019*), which will trigger cell rupture, accelerated oxidation reaction, leading to faster decline in quality (*Zhou et al., 2019*).

In the context of chemical preservation, the application of SD has been demonstrated to successfully reduce the respiration intensity of *V. volvacea* and promote the firmness of the mushroom following treatment with 0.04% SD, as shown in this study.

Chemical preservation uses SD to treat the mushroom, in this article, 0.04% SD treatment can effectively inhibit the respiration intensity of *V. volvacea* and make the mushroom hard. *Chen et al. (2019)* reported that treatment with 0.75 μL·L$^{-1}$1-methylcyclohexapropene can maintain the hardness of *V. volvacea* and delay the appearance of the peak of respiration. In addition, the activities of SOD and POD enzymes were higher than that of the control group during the storage period, because SOD enzyme can provide strong protection against reactive oxygen species, which is an important part of the defence system, and can maintain the balance of intracellular oxygen metabolism and delay the aging of the mushroom body (*Sun et al., 2023*). POD enzyme can effectively scavenge the free radicals *in vivo* and maintain the normal functioning of the organism, and plays an important role in the storage and preservation of mushrooms after harvest (*Zhong et al., 2023*). The higher the activity levels of both enzymes, the greater the significance of the aging process of the mushroom body (*Chen et al., 2019*). Therefore, the higher the activity of the two enzymes, the slower the aging of the mushroom. The respiration intensity and MDA variables were significantly reduced by 39.55% and 31.01%

compared to the control, and MDA is one of the main indicators for evaluating the degree of lipid peroxidation of cell membranes and studying the aging process of organisms, so the lower the MDA content, the longer the freshness of the mushrooms. In particular, the MDA content and the increase in membrane permeability in the late storage period were much lower than the control level, which is similar to the results of *Shi (2020)* and may delay the aging process of *V. volvacea* by inhibiting the process of membrane lipid peroxidation and maintaining the integrity of the membrane structure.

In this article, the combined treatment of soaking the mushroom of *V. volvacea* in sodium dehydrogenate solution followed by irradiation was adopted. The result showed that this combined approach had better preservation effect than that of the single treatment of sodium acetate or irradiation. After harvesting, all kinds of metabolic activities are still going on in the body, the water content of *V. volvacea* is large, so the mushroom respiration is vigorous, the higher respiration intensity accelerates the consumption of nutrients, which makes the *V. volvacea* less suitable for storage. In this study, the composite treatment significantly reduced the respiration rate by 35.07% compared with the control group; this result is consistent with (*Wei et al., 2022*) that the effect of treating *Agaricus bisporus* with combination solution of chitosan and ε-polylysine (6:4) reduced the decay of fruiting bodies. In this article, the hardness of the mushroom was increased by 81.19%, 97.96% and 168.81% when composite treatment was applied than in the control, B and C treatments, respectivel. This is in corroboration to the studies by *Jiang, Feng & Zheng (2012)*, that the application of composite chitosan and tea polyphenol, as well as chitosan and thyme oil, to *Lentinula edodes* yielded analogous preservation effects. These effects included the preservation of hardness, inhibition of respiration rate, and reduction of weight loss rate. Chitosan and cinnamon extract applied to *Hypsizygus marmoreus* (*Zhao, Mu & Li, 2016*) inhibited respiration intensity, delayed soluble solids and Vc degradation which is consistent with the result of our article in which the composite treatment increased Vc content by 20.28% compared to the control group. The composite treatment also significantly reduced browning of *V. volvacea*, which was similar to the application on *Coprinus comatus* (*Qiao, 2012*) with chitosan, garlic juice and ginger juice, which slowed down water dissipation, reduced the rate of weight loss, inhibited browning, respiratory intensity and PPO activity results were consistent. As storage progressed, the treatment involving both SOD and POD enzymes demonstrated a substantial rise in activity, which was significantly greater than that observed with other treatments.

The correlation analyses of the parameters indicated that, throughout the storage period, the indicators supporting the preservation of mushroom quality (such as hardness, antioxidant enzyme activity, Vc, and soluble protein content) were inversely related to indicators associated with quality degradation (including MDA content, browning rate, respiration rate, *etc.*). Notably, Vc content exhibited a significantly positive correlation with browning rate. Additionally, physiological indicators linked to mushroom harvesting displayed a discernible trend. Comprehensive analysis of the composite treatment of different edible mushroom species research reported, have shown that the composite treatment is more capable of achieving the effect of superimposed or synergistic

preservation. As a result, it facilitates a significant extension of the shelf life of the mushrooms.

In conclusion, the study demonstrated effective preservation of *V.volvacea* by using irradiation and sodium dehydrogenate as a preservative. However, there are still some problems that need to be resolved. Firstly, the sample size selected for this study is for a specific point in time and may vary from season to season. Secondly, the sensory evaluation of treated mushrooms is somewhat subjective and should be evaluated by more than one person and then combined. Finally, there is also a certain amount of extrusion loss during post-harvest preservation of *V.volvacea*, and the number of experimental samples should be increased to minimize experimental error in order to provide an experimental basis for delaying the post-harvest quality deterioration of *V.volvacea*.

## CONCLUSION

This study found that selecting the right combination of preservation treatments can have a synergistic effect to maximize the preservation effect. The 0.8 kGy$^{60}$Co$\gamma$ irradiation treatment alone had a greater increase in Vc content, and the 0.04% SD treatment alone had lower respiration intensity and MDA content than other treatments. The composite treatment had higher mushroom hardness, soluble protein content, SOD enzyme activity, POD enzyme activity, respiration intensity and browning rate were significantly lower than other treatments. The composite treatment was superior to the single treatment, which could effectively reduce the respiration rate, browning degree and MDA content, inhibit the degree of membrane peroxidation, inhibit or retard the physiological and biochemical reactions of the mushroom, increase the activity of peroxidase, reduce the loss of soluble proteins, reduce the degree of mushroom softness, improve the appearance and quality of the *V. volvacea* more over it was conducive to the freshness of the *V. volvacea* after harvesting, thus prolonging the shelf-life of the *V. volvacea*. Consequently, future investigations may explore the preservation impact of straw mushrooms in conjunction with alternative preservation methodologies. This approach can also be extended to various other mushroom varieties to identify post-harvest changes in energy levels, membrane peroxidation levels, and antioxidant enzyme activities, as well as to examine the mechanisms by which different treatments sustain mushroom quality and impede senescence. The food industry could potentially benefit from advancements in vegetable and mushroom preservation. Further research is essential to uncover the effective components and detailed mechanisms underlying the composite treatment.

## ACKNOWLEDGEMENTS

The authors would like to thank Dr. Qiuhui Hu, Donglu Fang for providing the facilities for analyses.

### Funding
This work was supported by the funding from the Jiangsu Agriculture Science and Technology Innovation Fund of Jiangsu Province (grant no. CX (22)3141). The funders had no role in study design, data collection and analysis, decision to publish, or preparation of the manuscript.

### Grant Disclosures
The following grant information was disclosed by the authors:
Jiangsu Agriculture Science and Technology Innovation: CX (22)3141.

### Competing Interests
The authors declare that they have no competing interests.

### Author Contributions
- Mingjuan Mao conceived and designed the experiments, performed the experiments, analyzed the data, authored or reviewed drafts of the article, and approved the final draft.
- Lin Ma performed the experiments, authored or reviewed drafts of the article, and approved the final draft.
- Ning Jiang performed the experiments, prepared figures and/or tables, and approved the final draft.
- Jinsheng Lin performed the experiments, prepared figures and/or tables, and approved the final draft.
- Shaoxuan Qu analyzed the data, prepared figures and/or tables, and approved the final draft.
- Huiping Li analyzed the data, prepared figures and/or tables, and approved the final draft.
- Ping Xu performed the experiments, analyzed the data, prepared figures and/or tables, and approved the final draft.
- Di Liu conceived and designed the experiments, analyzed the data, authored or reviewed drafts of the article, and approved the final draft.
- Lijuan Hou conceived and designed the experiments, performed the experiments, authored or reviewed drafts of the article, and approved the final draft.

### Data Availability
The raw measurements are available in the Supplemental File.

### Supplemental Information
Supplemental information for this article can be found online at http://dx.doi.org/10.7717/peerj.18177#supplemental-information.

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
