# Peer review of "Effects of 60Coγ-irradiation combined with sodium dehydrogenate on post-harvest preservation and physiological indices of Volvariella volvacea"

_PeerJ, doi:10.7717/peerj.18177_

## Round 0.1 · original submission · Major Revisions

If you feel you can revise your manuscript according to the reviewers' comments, please revise your manuscript and submit it. Please also send us your written responses to each of the reviewers' comments.

Yours,

Yoshi

Prof. Yoshinori Marunaka, M.D., Ph.D.

Reviewer 1 ·

Basic reporting

Abstract:
• Clarity and Conciseness: The abstract should be a clear, concise summary of the research. Ensure it includes the purpose, methodology, key results, and conclusions.
• Components: Make sure it covers the following components:
o Background/Purpose: Why was the study conducted?
o Methods: Briefly describe the methods used, including the type of radiation and dosage.
o Results: Summarize the key findings, such as the preservation outcomes and any observed changes in mushroom quality.
o Conclusions: What can be concluded from the study? How does it contribute to the field?
Introduction:
• Background Information: Provide context about the importance of mushroom preservation and current methods used.
• Research Gap: Highlight the gap in existing research that this study aims to fill.
• Objectives: Clearly state the research objectives. For instance, "This study investigates the efficacy of gamma radiation in preserving the quality and extending the shelf-life of white button mushrooms."
Materials and Methods:
• Detail: Provide detailed information on the materials and methods used. This should include:
o Sample Preparation: How were the mushrooms prepared for the study?
o Radiation Treatment: Specify the type and source of radiation, dosage levels, and exposure time.
o Assessment Methods: Describe how the preservation effectiveness was measured (e.g., physical appearance, microbial load, nutritional quality).
• Reproducibility: Ensure the methodology is detailed enough to allow other researchers to replicate the study.
Results:
• Presentation: Present the results clearly, using tables, graphs, and figures as necessary.
• Detail: Include specific findings such as changes in texture, color, and microbial counts post-radiation.
• Statistical Analysis: Provide information on the statistical methods used to analyze the data and include significance levels where appropriate.
Discussion:
• Interpretation: Interpret the results in the context of existing research. Discuss why certain results were obtained and their implications.
• Comparisons: Compare the findings with those from other preservation methods.
• Limitations: Acknowledge any limitations in the study, such as sample size or variations in radiation dosage.
Conclusions:
• Summary: Summarize the key findings of the research.
• Implications: Discuss the practical implications of the findings for the food industry and potential benefits for mushroom preservation.
• Future Research: Suggest areas for future research, such as exploring different types of mushrooms or radiation dosages.

Experimental design

good

Validity of the findings

good

Reviewer 2 ·

Basic reporting

In this paper, effects of three treatments on Volvariella volvacea were investigated. The topic is interesting and meets the scope of Peer J. The experiments were well organized, and results are reasonable. Thus, I recommend the paper for publication in Peer J after minor revisions.
1. Title: It is suggested to change 60Co withγ-irradiation. The title of the manuscript does not summarize the research content, and it is suggested to modify the topic title.
2. Abstract:
(1) There is lack of background and conclusion.
(2) Line 45: “B” should be revised to “C”.
(3) Line47-50: Repeat.
(4) Abbreviations should be spelled out at first use. Please check and modify similar issue in the manuscript.
(5) Line 57-59: The purpose of the article should be placed at the top of the abstract.
3. The manuscript mentioned that low-temperature storage (0-5 °C) is not suitable for preserving fresh mushrooms. It is well known that low temperatures are the main mode of storage for most mushrooms. Despite Volvariella volvacea is a type of high-temperature edible fungus that is prone to low-temperature autolysis,Therefore, authors should not generalize.
4. Line 109: What was the duration of transport and its condition (Temperature and RH)? It was not mentioned in experimental design.
5. Line113: Please provide specific time and dose of irradiation.
6. Line116: What exactly does the control group refer to?
7. Line 147: The correct title is “The Vc content and the soluble protein content assay” It was not same as the Line 141.
8. Line 409-411: Latin should be in italics.
9. In figures, both small and capital letters are used to denote significant differences. Similar format should be followed.
10. In all the figures, shelf life (d) is written along the X axis. It is 'days after storage (d)', not the shelf life.

Experimental design

The experiments were well organized

Validity of the findings

The results are reasonable.

Reviewer 3 ·

Basic reporting

The research on the effect of sodium dehydrogenate compounded with 60Co-irradiation on the preservation of Volvariella volvacea is interesting, but there are some suggestions in this manuscript:
1. The manuscript writing in general needs improvement in grammar so that the meaning in each sentence is clear. Professional proofreading of the manuscript should be done to improve the grammar.
2. The literature references used are from at least the last 10 years. In this manuscript, there are still many references that use more than 10 years of literature.
3. In Figure 2, the difference of each treatment should be marked so that it will be clearer the impact of each treatment on the results obtained.
4. In Figure 3 on the effect of different treatments on the hardness of V. volvacea, the unit of hardness should be included in the graph.

Experimental design

1. The research topic studied is in accordance with the scope of the journal.
2. The research gap is not clearly stated in the introduction.
3. The research method, especially how to determine the parameters used in the research, is not communicated in detail. For example, on line 126: the tools used, the process conditions used to analyze the hardness level are not conveyed. Line 129 also does not explain in detail how it is determined.

Validity of the findings

1. The introduction does not clearly state the novelty of the research nor the purpose of the research.
2. The problems presented in the introduction are not comprehensive enough.
3. The discussion section lacks a comprehensive and in-depth presentation of the data phenomena obtained in each analysis parameter. And also the relationship between the parameters used in this study.
4. The conclusion does not answer the objectives carried out. The research objectives are not specifically expressed in this manuscript.

---

## Round 0.2 · accepted · Accept

Congratulations again, and thank you for your submission.
Yours,
Yoshi
Prof. Yoshinori Marunaka, M.D., Ph.D.

Reviewer 2 ·

Basic reporting

no comment

Experimental design

no comment

Validity of the findings

no comment

Additional comments

All issues have been modified.